# Introducing a Novel Innovative Technique for the Recording and Interpretation of Dynamic Coronary Angiography

**DOI:** 10.3390/diagnostics14121282

**Published:** 2024-06-17

**Authors:** Thach Nguyen, Khiem Ngo, Tri Loc Vu, Hien Q. Nguyen, Dat H. Pham, Mihas Kodenchery, Marco Zuin, Gianluca Rigatelli, Aravinda Nanjundappa, Michael Gibson

**Affiliations:** 1Cardiovascular Research Laboratories, Methodist Hospital, Merrillville, IN 46410, USA; loc.vu@ttu.edu.vn (T.L.V.); nguyenquanghienmd@gmail.com (H.Q.N.); drmihasmamu@yahoo.com (M.K.); 2School of Medicine, Tan Tao University, Duc Hoa 82000, Long An, Vietnam; 3Department of Medicine, University of Texas Rio Grande Valley, Valley Baptist Medical Center, Harlingen, TX 78550, USA; luoikiemvang@gmail.com; 4Department of Medicine, Conemaugh Memorial Medical Center, Johnstown, PA 15905, USA; huandat123@gmail.com; 5Department of Translational Medicine, University of Ferrara, 44121 Ferrara, Italy; zuinml@yahoo.it; 6Interventional Cardiology Unit, Division of Cardiology, AULSS6 Ospedali Riuniti Padova Sud, 35043 Padova, Italy; jackyheart@libero.it; 7Peripheral Interventions, Cardiovascular Department, Cleveland Clinics Main Campus, Cleveland, OH 44195, USA; nanjuna@ccf.org; 8Baim Institute of Clinical Research, Harvard Medical School, Boston, MA 02115, USA

**Keywords:** coronary collision, dynamic coronary angiography, laminar flow, turbulent flow, water hammer shock, re-circulating flow, antegrade coronary flow, retrograde coronary flow

## Abstract

In the study of coronary artery disease (CAD), the mechanism of plaque formation and development is still an important subject for investigation. A limitation of current coronary angiography (CAG) is that it can only show static images of the narrowing of arterial channels without identifying the mechanism of the disease or predicting its progression or regression. To address this limitation, the CAG technique has been modified. The new approach emphasizes identifying and analyzing blood flow patterns, employing methodologies akin to those used by hydraulic engineers for fluid or gas movement through domestic or industrial pipes and pumps. With the new technique, various flow patterns and arterial phenomena—such as laminar, turbulent, antegrade, retrograde, and recirculating flow and potentially water hammer shock and vortex formation—are identified, recorded, and classified. These phenomena are then correlated with the presence of lesions at different locations within the coronary vasculature. The formation and growth of these lesions are explained from the perspective of fluid mechanics. As the pathophysiology of CAD and other cardiovascular conditions becomes clearer, new medical, surgical, and interventional treatments could be developed to reverse abnormal coronary flow dynamics and restore laminar flow, leading to improved clinical outcomes.

## 1. Introduction 

In the study of coronary artery disease (CAD), the precise mechanisms by which risk factors initiate and promote the progression of coronary lesions remain unclear. Furthermore, there is no consensus on why many lesions remain stable over extended periods, while others become unstable or acutely occluded [1,2,3]. To address this gap, our team has shifted focus from traditional biological studies and non-invasive imaging to employing fluid mechanics (FM) as the primary methodology for investigating healthy and diseased coronary arteries [4,5,6].

For FM engineers, laminar or helical flow is considered the ideal transportation modality in pipes because it maximizes efficiency and extends the life of equipment, while turbulent flow damages the inner surfaces of pipes and pump components. As a result, our research protocols have pivoted towards examining potential correlations between laminar flow, turbulent flow, and other arterial phenomena—including collisions and vortex formation—and their association with the presence, absence, progression, or regression of coronary plaques [7,8]. To elucidate the dynamic characteristics of coronary flows, our team has implemented a novel angiographic protocol, modifying the current conventional recording techniques [9,10,11]. 

In this article, first section provides a detailed description of the techniques used to accurately capture images of arterial flows and phenomena. The second section presents case studies that explore the potential correlations between coronary lesions and various flow patterns, elucidating the disease mechanisms from an FM perspective. Ultimately, the goal is to clarify the cardiovascular pathophysiology caused by flow disturbances, thereby enabling the development of innovative medical, surgical, and percutaneous interventions.

## 2. Dynamic Coronary Angiography

### 2.1. A New Technique in Dynamic Coronary Angiography

Currently, coronary angiography (CAG) involves filling the coronary artery with contrast to create a shadow of the lumen. This shadow is used to detect indentations, which are then identified as lesions or stenoses (Figure 1). However, this traditional method has a significant limitation: it only provides a static image of the narrowing in the arterial channel and does not offer insights into the disease mechanism or its potential progression or regression. To address this limitation, our team has developed a new technique by modifying the current angiographic recording sequence and reprogramming the reviewing protocol. This innovative approach allows for a more dynamic and comprehensive assessment of healthy and diseased coronary arteries.

The new dynamic angiographic technique aims to distinctly image and accurately identify various types of flows, such as laminar, helical, peripheral, antegrade, and retrograde flow, as well as arterial phenomena like vortex formation, water hammer shock, and turbulence. It is crucial to document the presence or absence of these flows and phenomena in the proximal segment, midsection, or distal region of a plaque. Additionally, the technique requires precise description and recording of the plaque’s border—whether it is clean and sharp or hazy—as these characteristics reflect the dynamic effects of normal or abnormal flows and the arterial phenomena on the plaque’s surface and endothelial layers.

By employing this new dynamic recording and reviewing technique, investigators can focus on identifying flows, analyzing their dynamics, and planning interventions. This approach adapts methodologies and practices commonly used by fluid mechanics or hydraulics engineers when transporting fluids or gases through residential or industrial pipes and pumps.

### 2.2. The Injection of Contrast and the Recording of Images

Initially, the contrast medium is injected until the index coronary artery is fully opacified. The manual injection stops when black-colored contrast is seen ejecting back from the coronary ostium into the aorta (Figure 2A). As blood enters the vessel, the shape, movements, directions, and interactions of the white-colored blood flow can be clearly observed and easily identified against the black contrast background (Figure 2B,C). The coronary angiogram is recorded from the start of the injection until all the contrast disappears from the distal arterial vasculature, marking the arterial phase (Figure 2D). The recording concludes after the contrast is flushed out of the coronary veins, marking the venous phase (Figure 3A–C).

During the recording, the camera is positioned at an angle that can record the index artery and vein at their full length, with all the images completely visible within the screen at 15 images per second (s) or 0.067 s or 67 milliseconds (ms) per image (67 ms is the interval between recorded images). 

## 3. Angiographic Views 

In the new dynamic angiographic recording, all the detailed movements of the blood (and contrast) need to be captured distinctly and identified accurately on the clean background of the lungs. The selected angles of the projector circumvent superimposing the arteriograms or venograms onto the bony structures of the spine or the myocardium filled with contrast at the end of the arterial phase and during the venous phase. The criteria of excellence when recording the angiogram are listed in Table 1. 

### 3.1. The Left Main Artery 

The left main artery (LM) needs to be delineated at full length so the entry flow from the aortic root and the hydraulic phenomena in the mid-segment and the distal bifurcation can be captured. The best views for the ostial LM and the mid-segment are (1) the anteroposterior (AP) caudal view (Figure 4A,B) and (2) the left anterior oblique (LAO) caudal (spider) view. The flows in the LM can clarify the mechanism of entry flow and the angle of attack, which injures the intima at a specific location in the LM, starts a lesion, promotes its growth (at the upper or lower border), or ruptures its cover and precipitates acute coronary syndrome (Figure 5A–D).

### 3.2. The Left Anterior Descending Artery

In the left anterior descending artery (LAD), the flow to be seen is in the ostium, the outer curve or the upper border of the proximal segment, which is a preferred location for plaques, an area with low shear stress and at the distal end of the proximal segment [9]. The best views for the ostial and mid-segment LAD are: The left anterior oblique (LAO) caudal (spider) view for the ostium (Figure 6A,B)The left anterior oblique (LAO) cranial view for the proximal segment (Figure 7A,B)The anteroposterior (AP) cranial view for the proximal and mid-segments (Figure 8A,B)

### 3.3. The Left Circumflex Artery

In the left circumflex artery (LCX), the flow to examine is in the ostium, the outer curve opposite to the carina, the mid-segment, and the obtuse marginal (OM) branches. The best views for the ostial and mid-segment LCX are (1) the AP caudal view and the right anterior oblique (RAO) caudal view, which show the ostium and the proximal segment of the LCX and the branches of the OM well. The AP caudal view is used when the proximal segment of the LCX is not tortuous. The right anterior oblique (RAO) caudal view is better when the proximal segment is curved or tortuous (Figure 9A,B).

### 3.4. The Right Coronary Artery

In the right coronary artery (RCA), the flows to look for are listed in Table 2. 

The best view for the flow in the ostial, proximal, and mid-RCA is the LAO caudal view (spider view) (Figure 10A). The LAO cranial view is the best view for flow in the distal RCA and bifurcation with the posterior descending artery and posterior lateral branch (Figure 10B).

### 3.5. Analysis Using Artificial Intelligence Programs

Extensive utilization of computational fluid dynamics studies, artificial intelligence, and deep learning packages and programs was initiated in our research protocols and aimed to optimize the research methodology on flows and arterial phenomena, systemize the protocol for angiographic interpretation, and give the preliminary online results without the need for offline manual image review [12,13,14,15,16,17].

## 4. Clinical and Imaging Correlations

In patients with cardiovascular (CV) disease, there are many clinical situations in which the pathophysiology is still not well elucidated. They are listed in Table 3. Can the new images obtained through dynamic coronary angiography, depicting both normal and abnormal arterial flows or phenomena, contribute to understanding the pathophysiological mechanisms of disease from an FM perspective? Additionally, could rectifying these disturbances in flow aid in proposing viable therapeutic solutions?

### 4.1. Correlation Case Study 1—Turbulence from Recirculating Flow and an Ostial Lesion in the Side Branch 

In the coronary angiograms of middle-aged patients, many have only subtle, minimal lesions in the ostium of the left circumflex artery (LCX) (Figure 11A). This is a slow-growth lesion, especially in patients with well-controlled hypertension and hyperlipidemia. Why are the lesions discovered first at this particular ostial LCX location when there are no visible lesions in other segments of the coronary vasculature? What is the FM explanation for this angiographic observation? From an FM perspective, when the fluid arrives at the bifurcation of a pipe, the fluid layers at the center of the divergent side branch (SB) exhibit a higher velocity compared to the peripheral layers adjacent to the outer wall (Figure 11B) [18].

When the velocity and pressure differences between the central and peripheral layers reach a critical threshold, the innermost layers detach from the boundary layer of the inner wall (Figure 12). This detachment causes the layers to curl upon themselves, resulting in the formation of recirculating flows and a stagnant area near the wall. Within this stagnant region, the flow velocity decreases, and pressure increases according to the Bernoulli equation. If the recirculating flows are sufficiently strong, they can evolve into vortices—swirling masses of fluid with rapid rotational motion. Ultimately, these vortices gradually decay into disorganized flow or turbulence (Figure 9A–D) [18].

The early development of lesions in the ostial LCX is attributed to this segment being confined within a narrow atrioventricular (AV) groove. Initially, as an atherosclerotic plaque forms, it gradually intrudes into the lumen, leading to flow obstruction. In response, the artery compensates by expanding, maintaining the lumen’s surface area despite the increasing plaque volume (Figure 13). However, because the proximal LCX is constrained within the AV groove, it lacks the capacity to expand. Consequently, lesions in the proximal segment of the LCX become noticeable earlier than those of a similar size in other coronary locations. 

In patients with lesions in the distal RCA, proximal to the bifurcation with a posterior descending artery, the mechanism of recirculating flow could be similarly applied. However, lesions form due to recirculating flow in the separation zone where the blood converges from the two branches in the retrograde direction (Figure 14A,B and Figure 15A–F).

### 4.2. Correlation Case Study 2—Coronary Lesions and Laminar versus Turbulent Flow

For patients with a moderate 50% stenotic lesion, some may become unstable unpredictably, while many others remain asymptomatic for an extended period [3]. No clinical nor testing criteria can reliably predict the progression nor regression of these lesions. How can this problem be addressed from an FM perspective? 

Using the new dynamic coronary angiographic technique, it has been observed that laminar flow is the most common flow pattern in coronary arteries (Figure 2C). According to FM principles, laminar flow protects the lining of pipes, the components of pumps, and the banks of waterways, whereas turbulent flow damages them [20,21]. Applying these FM principles to cardiovascular (CV) disease research, one can question whether turbulent flow at a coronary lesion further damages the plaque, ruptures its cap, and precipitates acute coronary syndrome (ACS). Conversely, could laminar flow at the lesion maintain the stability of the plaque cover and the integrity of the endothelial cell layer? If so, does laminar flow also inhibit plaque growth and sustain a patient’s stable angina (SA) condition? These questions were examined in the following study [22,23].

A group of patients admitted with unstable angina underwent a new dynamic angiogram. They were selected based on the presence of a single moderate lesion in a straight coronary segment, distant from any bifurcation. The angiographic risk factors evaluated were laminar flow versus disorganized or turbulent flow across the lesion. All patients received the optimal medical management to maintain their blood pressure below 120 mmHg and their low-density lipoprotein (LDL) levels below 75 mg/dL (Figure 16A–F). The clinical endpoints were persistent stable angina versus acute coronary syndrome (ACS) requiring percutaneous intervention.

The results showed that patients with baseline laminar flow remained stable, neither developing ACS nor requiring percutaneous or surgical intervention at the two-year follow-up (Figure 16A–F). In contrast, 83% of patients with baseline turbulent flow across a moderate-sized lesion developed ACS within six months. This pilot study suggests the protective effect of laminar flow and an increased risk of plaque rupture due to turbulent flow [23]. Large randomized studies are needed to confirm these observations.

### 4.3. Correlation Case Study 3—Retrograde Flow Causing Chest Pain and Sudden Death

Many patients with aortic stenosis (AS) and patent coronary arteries experience chest pain (CP) and sudden cardiac death (SCD). What is the pathophysiological mechanism underlying these two classic symptoms of AS? Similarly, patients with dilated cardiomyopathy or anomalous coronary artery from the opposite aortic sinus of Valsalva (ACAOS) also present with CP and SCD. How can the disease mechanisms in these conditions be explained from an FM perspective?

Under normal conditions, using the new dynamic angiographic technique, coronary blood flow is observed moving forward, faster during diastole and more slowly during systole, due to the higher-pressure gradient between the aortic root and the left ventricle during diastole. Conversely, retrograde flow is rare and brief. However, in extreme cases, such as patients with ACAOS combined with an inter-arterial course between the aorta and the pulmonary trunk, the proximal segment may prominently show retrograde flow of the black contrast, spilling into the aortic sinus. With the dynamic coronary angiography technique, the retrograde flow of the contrast in black and the antegrade flow of the blood in white can be simultaneously observed at the proximal and distal ends of the flow (Appendix A).

This slow antegrade and intermittent retrograde flow can also be observed in patients with severe uncontrolled hypertension (systolic blood pressure > 200 mmHg) without coronary lesions. In contrast, patients with dilated cardiomyopathy and patent coronary arteries exhibit a significant delay in antegrade flow without the presence of retrograde flow. Consequently, both slow antegrade and reverse coronary flows hinder the timely delivery of highly oxygenated blood to the myocardium, leading to significant functional ischemia despite the absence of obstructive coronary lesions [24,25,26].

### 4.4. Correlation Case Study 4—Collision at the Middle of the Right Coronary Artery

In patients with CAD, many have lesions in the mid-RCA. What alterations in flow dynamics could contribute to the development of lesions in the mid-RCA segment? In a patient with unstable angina and a severe lesion in the RCA, static images or videos may show antegrade flow, stagnation of the contrast at the transition between systole and diastole, or a collision between antegrade and retrograde flows. How can these arterial flow dynamics be explained from an FM perspective? Could this phenomenon be analogous to a water hammer shock event? (Figure 17A–F, Appendix A).

### 4.5. Fluid Mechanics: Water Hammer Shock in Pipes

Water hammer shock, or hydraulic shock, is a transient phenomenon in fluid flow systems, particularly in pipelines. It occurs when there is an abrupt change in flow velocity or direction, typically triggered by sudden valve closures or rapid pump start/stop actions [27]. This sudden change in flow velocity generates a pressure surge within the system, accompanied by the formation of pressure waves that propagate through the fluid at the speed of sound, exceeding the actual flow velocity. When these pressure waves encounter alterations in the pipe system, such as closed valves or bends, they reflect back into the pipe, leading to a subsequent pressure build-up. As these pressure waves reflect back and forth within the pipe system, they can amplify each other, resulting in a rapid increase in pressure—a phenomenon known as the water hammer effect. The sudden surge in pressure can exert significant stress on the pipe walls, potentially causing vibrations, noise, and damage to pipes, fittings, and other system components if not adequately controlled (Figure 18). 

### 4.6. Water Hammer Shock in Coronary Arteries

In a similar context, considering the aorta as a tank and the coronary artery as an outflow pipe, the forward movement of coronary blood resembles fluid drainage. During the initial phase of systole, the left ventricle (LV), acting as a proximal valve, contracts, causing the coronary blood to temporarily halt at the distal myocardium. Over time, this results in the accumulation of stationary blood. Simultaneously, the contraction of the LV compresses the myocardial capillary network, forcing the stationary blood to move backward and collide with the anterograde flow, particularly at the transition from diastole to systole. This collision induces turbulence, leading to injury to the intima and promoting the atherosclerotic process (Appendix A). Based on the sequence of images illustrating this collision, it suggests many similarities to a water hammer shock event in a pipe (Figure 17A–F).

### 4.7. Angiographic and Clinical Interventions

The collision between antegrade and retrograde flows, or water hammer events, is frequently observed in the mid-RCA, the distal end of the proximal segment of the LAD and LCX, and the iliac artery. This phenomenon occurs during the transition between diastole and systole, when retrograde systolic flow clashes with antegrade diastolic flow. This detail is crucial, as it suggests that the preferred location for coronary lesions is determined more by temporal factors (timing) than anatomical factors (location). Lesions develop wherever coronary flow reaches by the end of diastole, transitioning to left ventricular contraction in systole.

As turbulent flow is the most plausible cause of atherosclerosis, the benefit of stenting may lie in eliminating turbulent flow and restoring laminar flow (Appendix A). Well-controlled systolic blood pressure and a decreased rate of pressure rise (dp/dt) achieved through beta-blockers can attenuate the effects of retrograde flow and reduce turbulence at collision sites. This mechanism likely underlies the benefits of controlled hypertension and beta blockade in preventing coronary artery disease (CAD) and myocardial infarction [28].

## 5. Conclusions

The investigation of coronary artery disease (CAD) predominantly centers on understanding the mechanisms underlying plaque formation, progression, and regression. The introduction of novel techniques for recording and interpreting coronary angiography has facilitated the identification of diverse flow patterns and arterial phenomena within coronary arteries. These newly elucidated flow dynamics suggest that the formation and growth of coronary lesions are primarily attributed to disorganized flow or turbulence. Turbulent flow causes injury to the intima, initiating the development of small lesions and fostering their progression, particularly in the presence of abundant low-density lipoprotein (LDL) cholesterol molecules. Uncontrolled hypertension exacerbates the repetitive impact of turbulent flow, increasing the likelihood of plaque rupture. This pathological mechanism is implicated in precipitating acute coronary syndrome, ST-elevation myocardial infarction, critical limb ischemia, transient ischemic attack, stroke, and other related conditions. The adoption of this innovative approach to recording and reviewing coronary dynamic flows, along with their clinical implications, has ushered in a new era of valuable insights and innovative applications for the diagnosis and medical, surgical, and interventional management of patients with CAD.

## Figures and Tables

**Figure 1 diagnostics-14-01282-f001:**
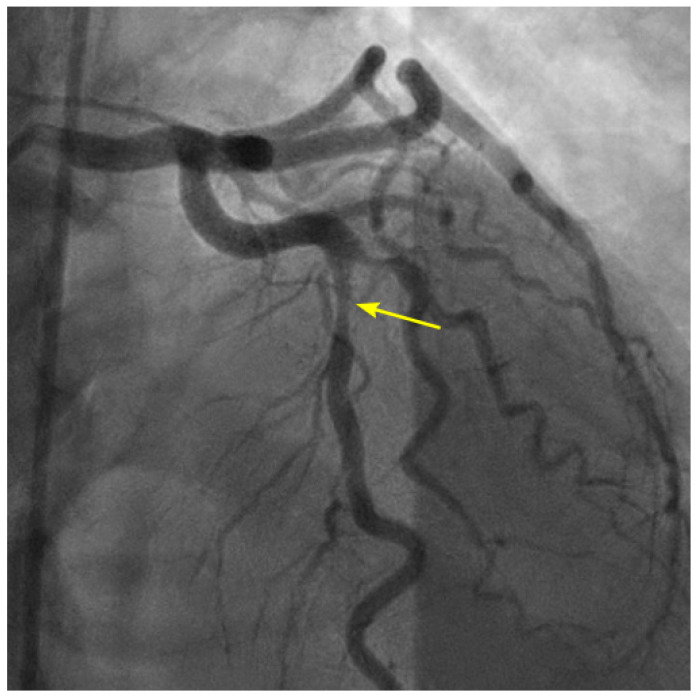
The left anterior descending artery (LAD) in the anterior–posterior cranial view. There is a narrowing in the mid-segment of the LAD, distal to the origin of a moderate-size diagonal (arrow). The current angiography technique only shows the degree of stenosis without critical information on how the lesion was formed and how it will progress in the near or far future.

**Figure 2 diagnostics-14-01282-f002:**
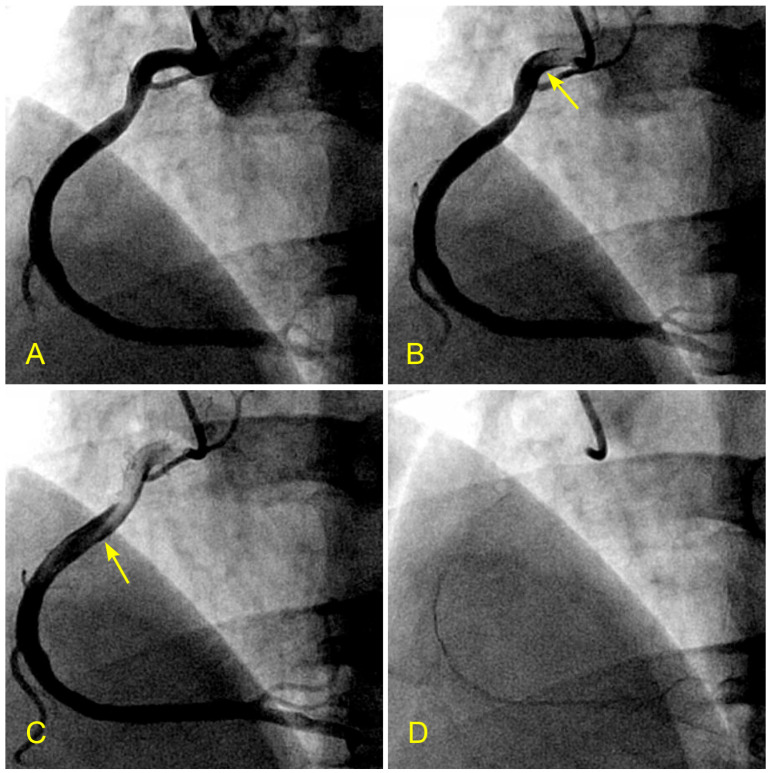
(**A**–**D**) Arterial phase. These images are not in consecutive sequence. (**A**) The artery is full of contrast in black. (**B**) The blood, in white, begins to move in (arrow). (**C**) The blood, in white, moves to the mid-segment. The tip of the blood flow is pointed, which is typical for laminar flow (arrow). (**D**) The contrast in black is almost washed out of the artery. This is the end of the arterial phase.

**Figure 3 diagnostics-14-01282-f003:**
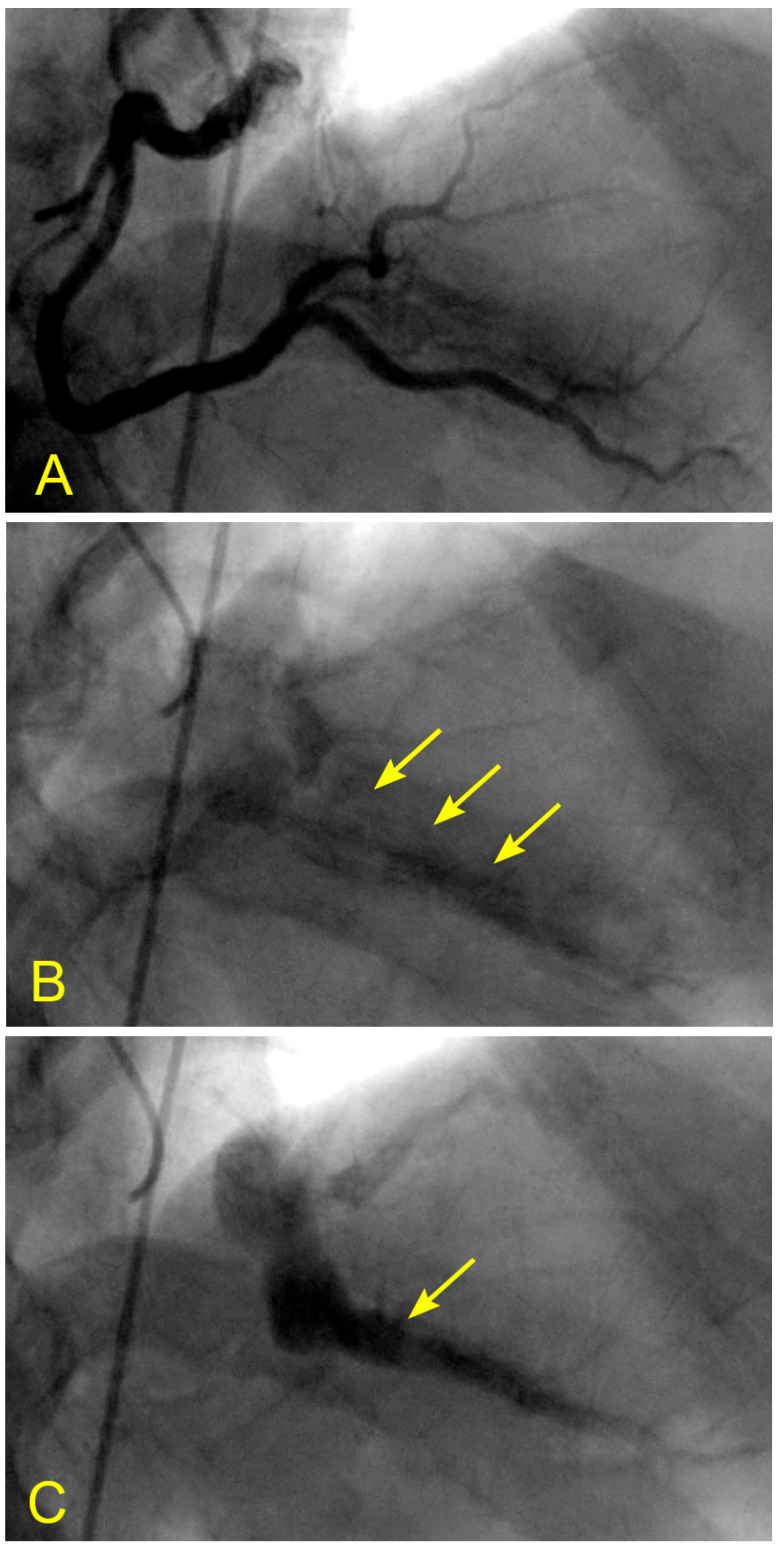
(**A**–**C**) The coronary venous flow. These images are not in consecutive sequence. (**A**) The contrast fills the right coronary artery. (**B**) Contrast goes into the myocardium (arrows) and coronary veins. (**C**) The contrast fills up the coronary vein (arrow).

**Figure 4 diagnostics-14-01282-f004:**
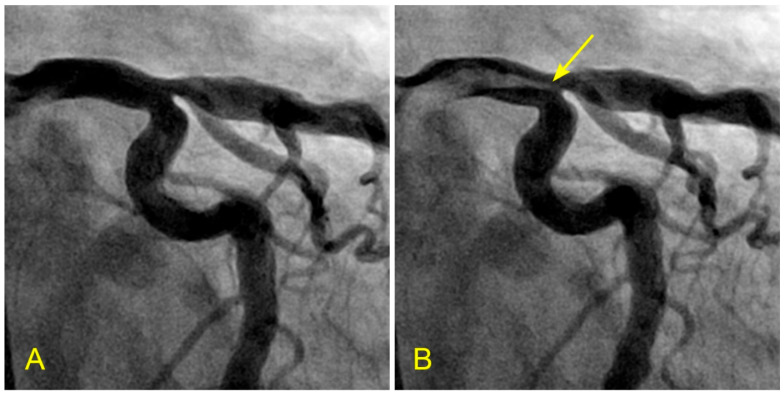
(**A**,**B**) The anteroposterior (AP) caudal view. These two images are not in consecutive sequence. (**A**) The left main coronary artery (LM) was seen well in terms of the bifurcation of the left anterior descending artery (LAD) and left circumflex artery (LCX). There was a narrowing in the ostium of the LAD. (**B**) The blood, in white, was seen moving in a laminar pattern with a pointed tip (arrow). The boundary layers on both sides of the laminar flow were well delineated. Even though the stenosis was severe in measuring the minimum lumen area (MLA) (>75%), the fractional flow reserve was negative. This patient has been stable for the last five years. The mechanism of clinical stability was most likely due to the presence of laminar flow across the lesion and the treatment with beta-blockers.

**Figure 5 diagnostics-14-01282-f005:**
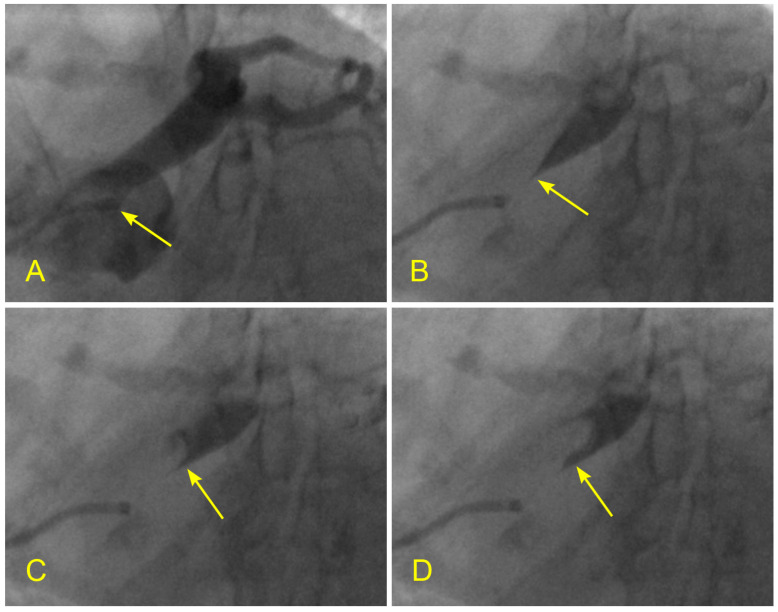
(**A**–**D**) Entry flow and angle of attack in the left anterior oblique (LAO) caudal view. These images are not in consecutive sequence. (**A**) The left main coronary artery (LM) was seen well in terms of its proximal and mid-segments. The tip of the catheter was outside the LM (arrow). (**B**) Because the LM was large, while the left anterior descending artery was much smaller, the contrast formed a thick boundary layer, which started in the mid-segment of the LM and became thicker at the distal end of the LM, in the form of a triangle (arrow). (**C**) At the beginning of diastole, the blood, in white, moved in and first hit the boundary layer at its base (arrow). (**D**) Then, the blood, in white, began to move up toward the center of the flow (arrow). Pictures (**C**,**D**) are important because the blood first hit the lower border of the LM. If there had been a lesion in the mid-LM, the jet of blood would have hit the base of the plaque and ruptured its cover at the junction between the cover and the normal intima. This is the mechanism of injury at a specific location in the LM based on the angle of attack.

**Figure 6 diagnostics-14-01282-f006:**
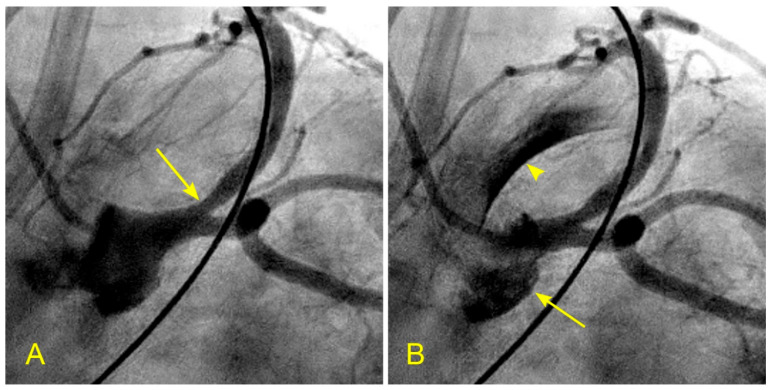
(**A**,**B**) Timing of systole and diastole in the left anterior oblique (LAO) caudal (spider) view. These are the two consecutive images of the left main artery (LM) in the spider view. They are separated by 0.06 s (recorded at 15 frames per second). (**A**) In this view, the ostium of the left anterior descending artery (LAD) is seen well, so the flow through it would be imaged well (arrow). (**B**). In this view, the contrast in black color is visibly ejected from the aortic root and the coronary sinus into the ascending aorta (arrow), with black contrast in the ascending aorta (arrowhead). These features help to time and differentiate between the flow in the coronary artery during systole and diastole.

**Figure 7 diagnostics-14-01282-f007:**
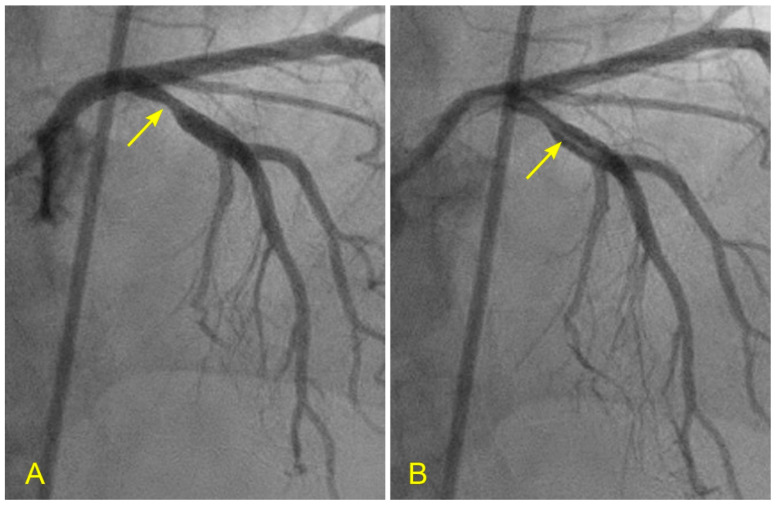
(**A**,**B**) Laminar flow across a lesion in the left anterior oblique (LAO) cranial view, which is important for the presence of antegrade laminar flow or turbulent retrograde flow. (**A**) In this anterior–posterior (AP) cranial view, the proximal segment of the left anterior descending artery (LAD) was completely filled with contrast. There was a moderate lesion (arrow). (**B**) In this view, the blood, in white, was seen to cross the stenotic segment in laminar flow (arrow) over a background of black contrast. Under the protective effect of laminar flow, the lesion stayed stable, not progressing to acute coronary syndrome while the patient was on optimal dose of beta-blockers and statin.

**Figure 8 diagnostics-14-01282-f008:**
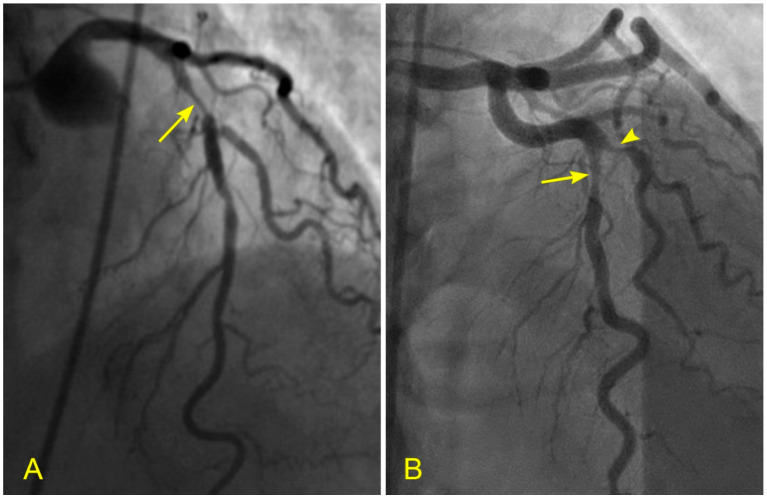
(**A**,**B**) Lesion in the proximal and mid left anterior descending artery (LAD) in the anteroposterior (AP) cranial view. This view is essential for searching for antegrade laminar or turbulent retrograde flow. (**A**) In this picture, there is a critical narrowing in the proximal segment of the LAD, proximal to the origin of a moderate-size diagonal (arrow). In our research protocol, for a lesion at this location, we would ask why and how the lesion formed in the proximal LAD, proximal to the ostium of the diagonal and the main branch (and not at other locations). Which fluid mechanics mechanism(s) was (were) responsible for forming lesions at this location? Could it have been due to retrograde flow from the diagonal and distal LAD? (**B**). This picture shows a critical narrowing in the mid-segment of the LAD (arrow) and the ostium of the moderate-size diagonal (arrowhead). In our research protocol, we would ask the same questions about the formation of lesions at these two locations.

**Figure 9 diagnostics-14-01282-f009:**
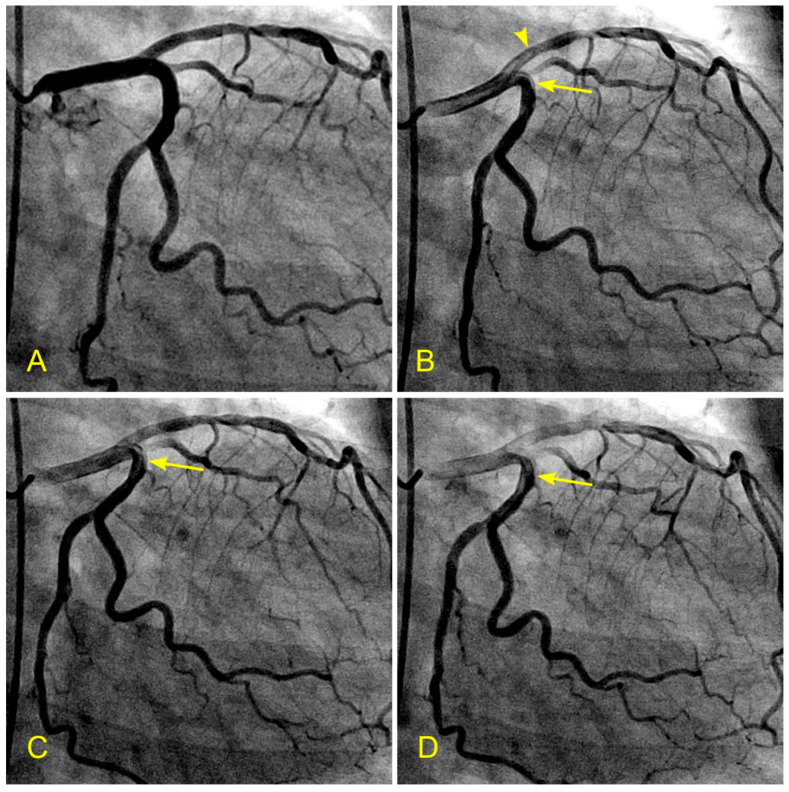
(**A**,**B**) Recirculating flow in the anterior–posterior (AP) caudal view. These are pictures of the left main artery (LM) and the left circumflex artery (LCX). (**A**) Both arteries are filled with contrast. (**B**,**C**) The blood, in white, moves in; however, the blood fills the LAD first (arrowhead) and then, 0.06 s later, turns the corner and moves into the LCX. The blood, in white, hits the inner curve on the carina side first (arrow). (**D**) First, the central layers flow faster than the layers at the border. If the difference or gradient between the speed of these layers reaches a critical point, the peripheral layers will twist on themselves (arrow). A vortex may be formed if the flow speed is high enough, and turbulent flow may follow (arrow). These abnormal flows may damage the intima and start the atherosclerotic process [10].

**Figure 10 diagnostics-14-01282-f010:**
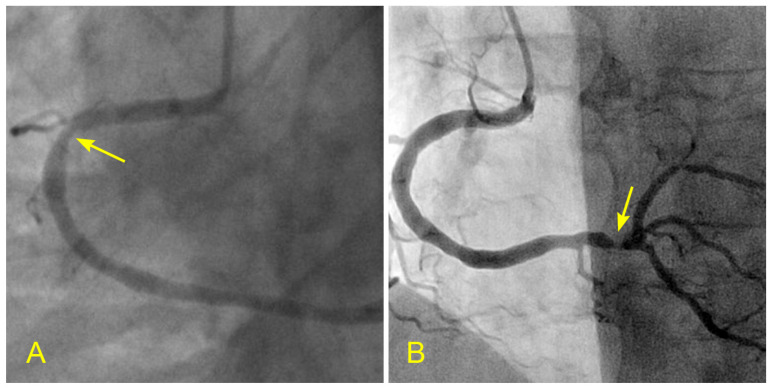
(**A**) Vulnerable plaque in the anterior–posterior (AP) caudal view. The right coronary artery (RCA) is filled with contrast. There is a lesion at the junction between the proximal and mid-segment, mainly at the inner curve of the RCA (arrow). The cover of the plaque is seen to be open at the proximal end, at the junction between the plaque and the normal intima. The lesion is most likely caused by repetitive movement at a hinge. The location of the injury (proximal border of the plaque) is most likely due to recurrent hits from the antegrade flow, correctly targeted from an appropriate angle of attack (please see also Figure 5C,D). (**B**) This is a lesion in the distal segment of the RCA (arrow). Why was the lesion formed proximal to the bifurcation with the posterior descending artery rather than distal to it? Could it be due to recirculating flow due to flow from two converging branches in a retrograde direction?

**Figure 11 diagnostics-14-01282-f011:**
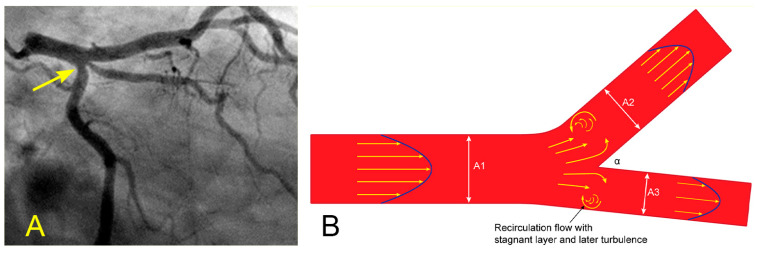
(**A**,**B**) Ostial lesion and diverting flows at a bifurcation. (**A**) In an angiogram of a middle-aged woman, all the arteries are patent except for a minimal subtle narrowing in the ostium of the left circumflex artery (arrow). (**B**) When the differences in velocity between the central and peripheral layers reach a critical level, the flow at the peripheral layers recirculates.

**Figure 12 diagnostics-14-01282-f012:**
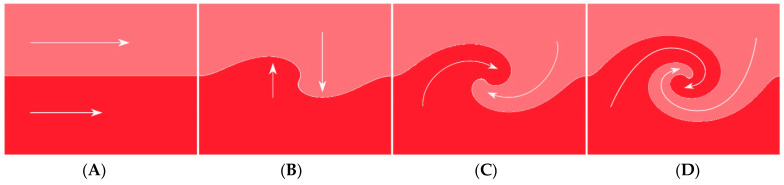
Layer separation and recirculation. (**A**) Schematic representation of flows with different velocities entering a curved slope at fast speed in the central layers (where the pressure is lower). (**B**,**C**) Because the peripheral flows move from a high-pressure (with a lower speed) to a lower-pressure location at the central flows, the innermost layers of the peripheral flows are entrained and pulled into the central flow. (**D**) Separation of layers and subsequent creation of recirculating flow (Adapted from reference [18]).

**Figure 13 diagnostics-14-01282-f013:**
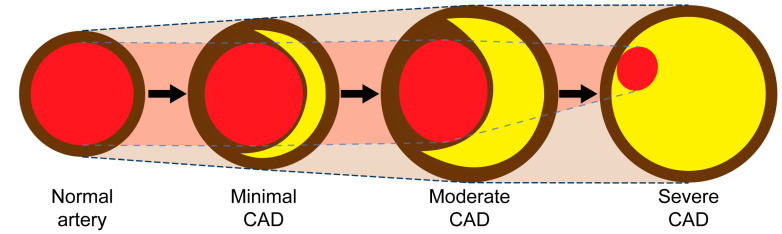
Compensatory expansion of the coronary artery so the surface area of the lumen (in red color) stays unchanged even when the plaque volume (in yellow color) increases (Adapted from reference [19]).

**Figure 14 diagnostics-14-01282-f014:**
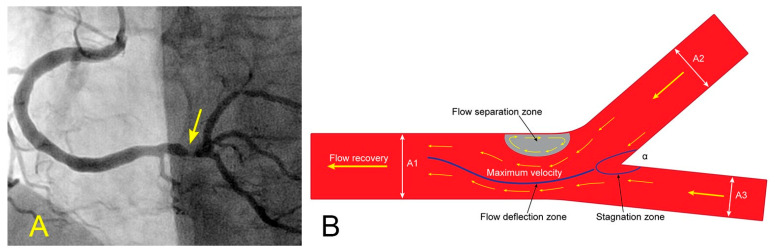
(**A**,**B**) Recirculating flow due to blood converging from two branches in the retrograde direction. (**A**) Coronary angiogram of an elderly patient with a severe lesion proximal to the bifurcation of the distal right coronary artery (RCA) and the posterior descending artery (arrow). (**B**) The blood converges from a large main branch and one smaller side branch (The yellow arrows point to the retrograde direction of the blood flow). Because of differences in velocity, the flow at the inner curve recirculates and starts a slow atherosclerotic process [10].

**Figure 15 diagnostics-14-01282-f015:**
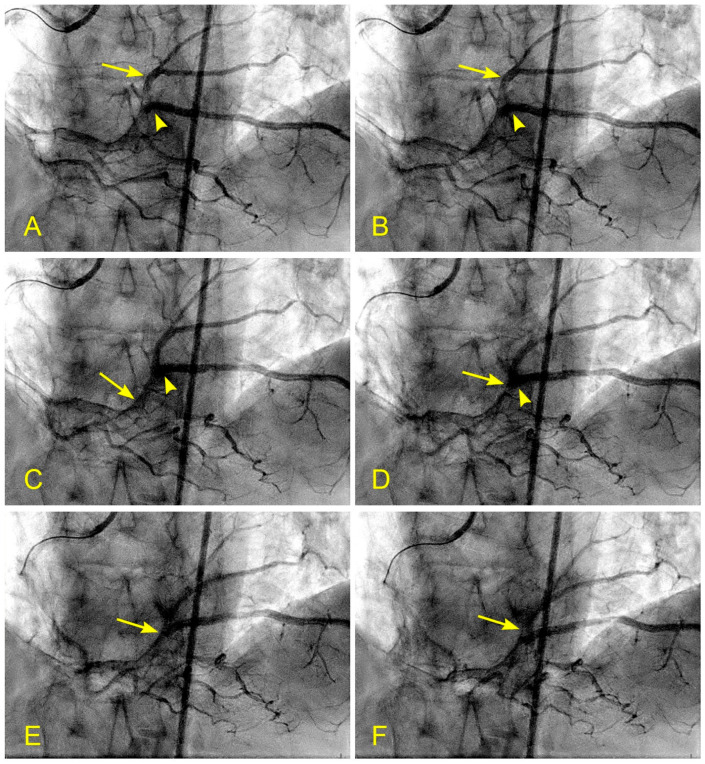
(**A**–**F**) Reverse flow in distal right coronary artery. This is a series of six sequential images separated by 6 milliseconds each (15 images per second). (**A**) The blood, in white, moves forward to the distal right coronary artery (RCA) (arrow) past the origin of the posterior descending artery (PDA) (arrowhead). (**B**) The blood (white) is now clearly distal to the origin of the PDA (arrow), while the contrast at the origin of the PDA stays stagnant and homogenously black (arrowhead). (**C**) Flow reversal. At the beginning of systole, at the distal RCA, the contrast reverses its direction and flows back past the origin of the PDA (arrow). (**D**) At the distal RCA, the blood (in white) pushes back the contrast (arrow) in the antegrade direction. The flow reversal is short-lived. (**E**,**F**) At the distal RCA, the blood, in white, moves forward as usual (arrow). If the reversed flow had been strong and lasted longer, more damage could have been inflicted on the endothelium and could have triggered the atherosclerotic cascade.

**Figure 16 diagnostics-14-01282-f016:**
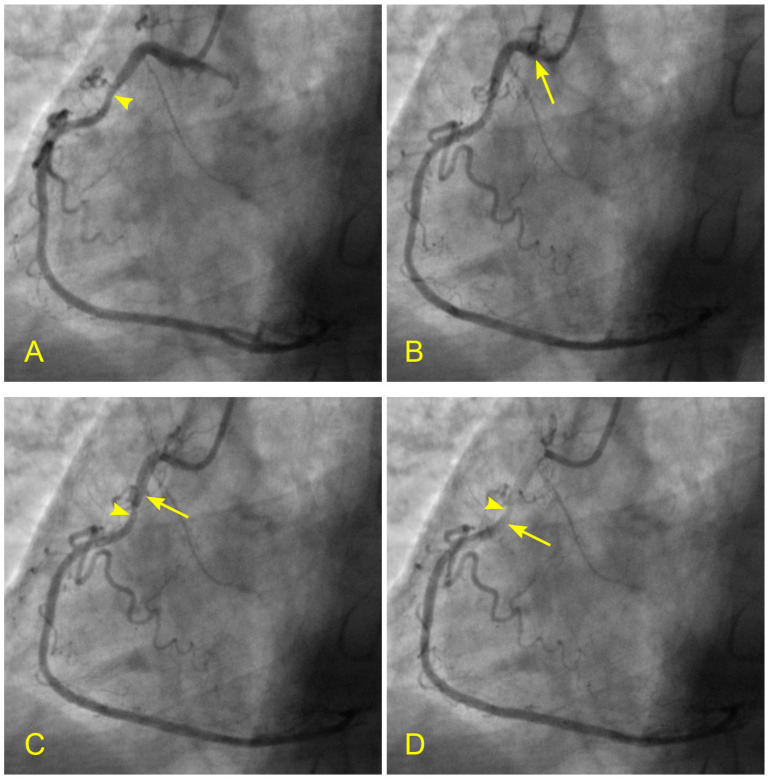
(**A**,**B**) Blood flow across a coronary lesion. These images are in continuous sequence. A middle-aged patient presented with unstable angina and underwent a coronary angiogram. (**A**). The artery was full of contrast in black. There was a 50% lesion in the mid-segment of the right coronary artery (RCA) (arrowhead). (**B**) At the ostium of the RCA, the blood, in white, moved in at the beginning of diastole (arrow). (**C**,**D**) The blood flow across a coronary lesion. (**C**) The blood, in white, was now at the beginning of the RCA mid-segment (arrow) (**D**). The flow of white blood passed the lesion section quickly without disorganized flow (turbulent flow) (arrow). The location of the lesion is marked with an arrowhead. (**E**,**F**) The blood flow across a coronary lesion. (**E**) The blood, in white, was now at the end of the RCA mid-segment (arrow). There was no reversed flow at the lesion site (arrowhead). (**F**) The flow of white blood entered the distal segment without disorganized flow (turbulent flow, arrow) at the lesion site (arrowhead).

**Figure 17 diagnostics-14-01282-f017:**
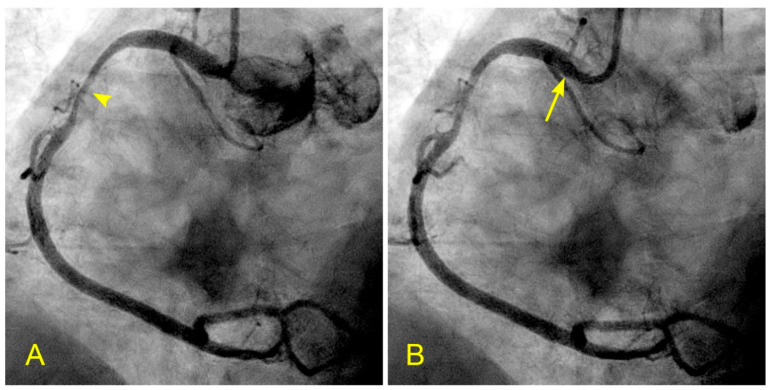
(**A**,**B**) Antegrade flow in the right coronary artery (RCA). These images are in continuous sequence. (**A**) The artery is completely full of contrast in black. There is a severe 80% lesion in the mid-segment of the RCA (arrowhead). (**B**) In the ostium of the RCA, the blood, in white, moves in at the beginning of diastole (arrow). (**C**–**F**) Antegrade and retrograde flows in the right coronary artery (RCA). (**C**) Now, the blood, in white, continues to move in at the proximal segment of the RCA (arrow). (**D**) The blood, in white, reaches the center of the mid-segment where the lesion is located (arrowhead). The contrast looks darker. This is the interface location between the antegrade and retrograde flow at the transition from diastole to systole. (**E**) The blood, in white, reaches the beginning of the distal part of the mid-segment (arrow). At this location of the transition between systole and diastole, the contrast still looks dark (arrowhead). (**F**) The contrast looks darker in the mid-segment, and in the proximal segment, the contrast in black looks darker and is at a standstill (red arrow). (**G**,**H**) Antegrade and retrograde flow in the right coronary artery (RCA). (**G**) The blood, in white, reaches the beginning of the distal segment (arrow). At the location of the transition between systole and diastole, the contrast still looks dark (arrowhead). The contrast looks darker in the proximal segment, where the contrast in black is at a standstill (red arrow). (**H**) The blood, in white, reaches the middle of the distal segment (arrow). At the location of the transition between systole and diastole, the contrast still looks less dark (arrowhead). The contrast looks lighter in the proximal segment (red arrow).

**Figure 18 diagnostics-14-01282-f018:**
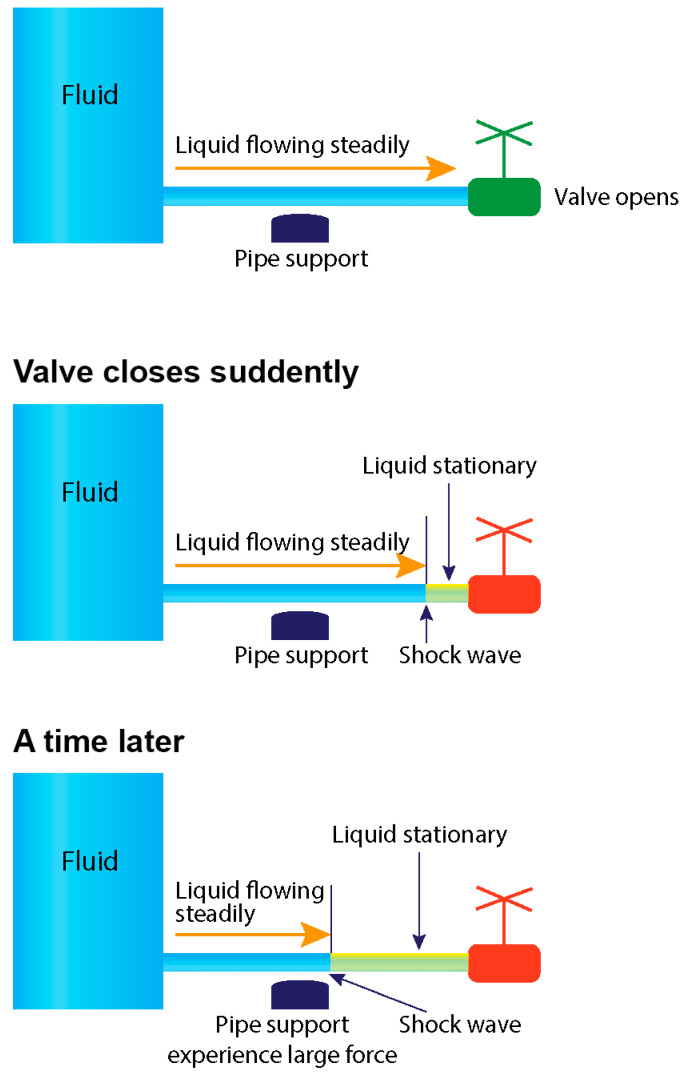
Water hammer event. The sequence of antegrade and retrograde flows results in collision and shock waves when the distal valve of a pipe is abruptly closed. Water hammer shock occurs when there is an abrupt change in velocity or flow direction in pipe systems, such as power failure, main breaks, pump start-up and shut-down operations, check-valve slam, rapid demand variation, and opening and closing of fire hydrants, as a pressure wave propagates in the pipe (Adapted from reference [27]).

**Table 1 diagnostics-14-01282-t001:** Criteria of angiographic excellence.

1 The left main artery (LM) needs to be delineated in its entire length from the ostium to the bifurcation in order to capture all the fluid mechanics (FM) mechanisms of entry flow and angles of attack, which affect the formation of lesions at the upper versus lower border, in the middle or distal LM. 2 The transition from the LM to the left anterior descending artery (LAD) and left circumflex artery (LCX) has to be delineated so that all the mechanisms affecting bifurcated flows and lesions (laminar or turbulent, at the center or the side, at the entry and exit shoulders of the LAD and LCX) can be identified, timed, and recorded. 3 The proximal segments of the LAD and LCX must be seen clearly, mainly the outer curve of the proximal LCX and LAD. Usually, there is a boundary layer with recirculating flow, where plaques are commonly formed. 4 The bifurcations of the LAD with the diagonal and the LCX with the obtuse marginal artery need to be delineated well to detect the antegrade or retrograde flow or collision causing LAD and LCX lesions proximal to the bifurcation. 5 The mid-segment of the right coronary artery (RCA) is where most lesions are located. Here is the location of a collision between antegrade and retrograde flows, which possibly inflicts the initial injury and triggers the atherosclerotic process. 6 The distal segment of the RCA at the junction between the RCA and the posterior descending artery (PDA) and the posterior lateral branch (PLB), where the majority of distal and slow-growth lesions are located. 7 The junction between the proximal and mid-segments of the RCA (or any curved segment) where lesions could develop due to the repeated hinge motion of the angle connecting the two arterial segments.

**Table 2 diagnostics-14-01282-t002:** List of locations to focus on in the right coronary artery.

1 The ostium, its orientation, and the angle formed by the proximal segment of the artery with the aortic wall (to investigate the mechanism of ostial lesions) 2 The angle connecting the proximal to the mid-segment (to check the hinge motion) 3 The mid-segment (to check the collision between antegrade and retrograde flows: most likely due to the water hammer shock phenomenon) 4 The junction between the mid- and distal segments (to check the hinge motion) 5 The bifurcation with the posterior descending artery and the posterior lateral branch (to check the recirculating flow caused by blood from convergent branches in a retrograde direction)

**Table 3 diagnostics-14-01282-t003:** List of cardiovascular conditions with unexplained pathophysiology.

1 Chest pain and sudden death in patients with aortic stenosis (AS) and patent coronary arteries. 2 Chest pain and sudden death in patients with anomalous coronary artery from the opposite aortic sinus of Valsalva (ACAOS) and patent coronary arteries. 3 Chest pain and sudden death in patients with dilated cardiomyopathy and patent coronary arteries. 4 Progression or regression of coronary lesion of moderate stenosis. 5 What is the mechanism of the formation and growth of lesions in the mid-segment of the right coronary artery? Could it be the water hammer shock phenomenon? 6 What is the mechanism of the formation and growth of lesions at the outer border of the ostium of the left circumflex? Could it be vortex formation from recirculating flow? 7 What is the mechanism of the benefit of stenting? Could it be due to restoring laminar flow? 8 In patients with new dilated cardiomyopathy and a low ejection fraction (EF), could a normal coronary dynamic flow predict the return to a normal EF with treatment?

## Data Availability

The original contributions presented in the study are included in the article/Appendix A; further inquiries can be directed to the corresponding author.

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
