# Peer review of "Introducing a Novel Innovative Technique for the Recording and Interpretation of Dynamic Coronary Angiography"

_diagnostics, 2024, doi:10.3390/diagnostics14121282_

Round 1
Reviewer 1 Report
Comments and Suggestions for Authors
Thank you for the opportunity to revise your study.
The study explores the fluid mechanics in CAD with potential to expand the findings also to PAD. One clinical benefit could be the tailoring of bypass anastomosis to prevent miointimal hyperplasia.
Overall the study presents a novel insight in the physiopathology of the CAD and proposes inovative diagnosis methods
Author Response
Please see

Reviewer 2 Report
Comments and Suggestions for Authors
The authors present a manuscript entitled “Introducing a New Innovative Technique of Recording and Interpretation of Dynamic Coronary Angiography”.
In the manuscript, the authors propose to modify the approach to describing inavasive coronary angiography by adding an in-depth description of the different patterns of blood flow in the coronary arteries. According to the authors' opinion, which they explain quite reasonably in the manuscript, this will improve the prediction of the course of coronary atherosclerosis and the understanding of the mechanisms of development of different clinical scenarios. In the manuscript, the authors describe the proposed criteria for assessing the quality of CAG performance in detail and with extensive illustrative material.
The authors further detail various clinical scenarios in which they demonstrate the description of possible mechanisms for the development of different clinical conditions using the proposed approach.
Overall, the manuscript appears to be a completed paper in which the authors argue the merits of the proposed approach to describe haemodynamic abnormalities in CAG in more detail. The manuscript has a good scientific soundness. I don't have any major comments on the manuscript.
Author Response
Please see
